# Optimizing mechanical ventilation: Personalizing mechanical power to reduce ICU mortality - a retrospective cohort study

**Ahmed S. Alkhalifah**[1]*, **Kenny Rumindo**[2], **Edgar Brincat**[3,4], **Florian Blanchard**[5], **Johan Helleberg**[6], **David Clarke**[7], **Benjamin Popoff**[8], **Olivier Duranteau**[9,10], **Zubair Umer Mohamed**[11,12], **Abdelrahman Senosy**[13]

**1** Pediatric Critical Care Unit, Qatif Central Hospital, Dammam, Saudi Arabia, **2** Research & Development, Getinge Acute Care Therapies, Solna, Sweden, **3** Pediatric Intensive Care Unit, Royal Hospital for Children, Glasgow, United Kingdom, **4** School of Medicine, Dentistry & Nursing, University of Glasgow, Glasgow, United Kingdom, **5** Department of Anesthesiology and Critical Care, Sorbonne University, GRC 29, AP-HP, DMU DREAM, Pitié-Salpêtrière Hôpital, Paris, France, **6** Department of Perioperative Medicine and Intensive Care, Karolinska University Hospital, Stockholm, Sweden, **7** Oxford Critical Care, Oxford University Healthcare Trust, Oxford, United Kingdom, **8** Department of Anesthesiology and Critical Care, Rouen University Hospital, Rouen, France, **9** Anesthesiology Department, Erasmus Hospital, Brussels, Belgium, **10** Intensive Care Unit, Hôpital d'instruction des Armées Percy, Clamart, France, **11** Adult Critical Care Unit, King Faisal Specialist Hospital and Research Centre, Madinah, Saudi Arabia, **12** Department of Anaesthesia and Critical Care, Amrita Institute of Medical Sciences and Research Centre, Kochi, India, **13** Adult Intensive Care Unit, Hayat National Hospital, Medina, Saudi Arabia

* ahmedsk2@gmail.com

**Data Availability Statement:** The data supporting the findings of this study consist of deidentified patient data sourced from the AmsterdamUMCdb,

## Abstract

### Background

Mechanical ventilation, a crucial intervention for acute respiratory distress syndrome (ARDS), can lead to ventilator-induced lung injury (VILI). This study focuses on individualizing mechanical power (MP) in mechanically ventilated patients to minimize VILI and reduce ICU mortality.

### Methods

A retrospective analysis was conducted using the Amsterdam University Medical Centers Database (AmsterdamUMCdb) data. The study included patients aged 18 and older who needed at least 48 hours of pressure-controlled mechanical ventilation. Patients who died or were extubated within 48 hours and those with inadequate data were excluded. Patients were categorized into hypoxemia groups based on their PaO2/FiO2 ratio. MP was calculated using a surrogate formula and normalized to ideal body weight (IBW). Statistical analyses and machine learning models, including logistic regression and random forest, were used to predict ICU mortality and establish safe upper limits for IBW-adjusted MP.

### Results

Out of 23,106 admissions, 2,338 met the criteria. Nonsurvivors had a significantly higher time-weighted average MP (TWA-MP) than survivors. Safe upper limits for IBW-adjusted

a database managed by the Amsterdam UMC. This dataset is available to qualified researchers after submitting an application and taking a training course. The AmsterdamUMCdb can be accessed at https://amsterdammedicaldatascience.nl/amsterdamumcdb/. The AmsterdamUMCdb provides a comprehensive repository of patient data that has been anonymized to protect patient confidentiality while enabling scientific analysis and validation of the study results.

**Funding:** The author(s) received no specific funding for this work.

**Competing interests:** The authors have declared that no competing interests exist.

MP varied across hypoxemia groups. The XGBoost model showed the highest predictive accuracy for ICU mortality. An individualization method for mechanical ventilation settings, based on real-time physiological variables, demonstrated reduced predicted mortality in a subset of patients.

## Discussion

Elevated TWA-MP is associated with increased ICU mortality, underscoring the need for personalized mechanical ventilation strategies. The study highlights the complexity of VILI and the multifactorial nature of ICU mortality. Further studies to define a safe upper limit for IBW-adjusted MP may help clinicians optimize mechanical ventilation settings and decrease the risk of VILI and mortality.

## Conclusions

Despite the fact that the study's retrospective design and reliance on a single-center database may limit the generalizability of findings, this study offers valuable insights into the relationship between mechanical power and ICU mortality, emphasizing the need for individualized mechanical ventilation strategies. The findings suggest a potential for more personalized, data-driven approach in managing mechanically ventilated patients, which could improve patient outcomes in critical care settings.

## Background

Although the definition of acute respiratory distress syndrome (ARDS) has evolved over the years, it encompasses the inflammation of lung tissue leading to deterioration of oxygenation and/or ventilation [1–3]. Mechanical ventilatory support (invasive or noninvasive) needed to treat ARDS could result in inadvertent harm, a phenomenon called ventilator-induced lung injury (VILI) [4]. The ventilator-generated causes of VILI include the pressures, volume, flow, and respiratory rate. The lung conditions favoring VILI depend on the amount of edema, which leads to decreased lung dimensions, increased lung inhomogeneity, and cyclic alveolar collapse and expansion. When a breath is delivered by the ventilator to the patient's respiratory system, some energy is used to overcome the resistance of the airways and expand the thoracic wall. A fraction of this energy is absorbed by the lung tissue, some of which leads to inflammation of the lung tissue [4,5].

Mechanical power (MP) is a measure of the amount of energy the ventilator delivers to the lung tissue and is a function of the ventilator settings, including tidal volume, pressures, flow, and respiratory rate. It is expressed in Joules/minute [5]. Not all the elements of MP have equal weight: doubling the VT leads to a fourfold increase in MP, doubling RR leads to a 1.4-fold increase, and doubling PEEP leads to a twofold increase in MP [5]. Animal experiments have shown a safe MP threshold to vary between 3 and 12 Joules/minute [6].

An association of high MP of ventilation with mortality has been established in ARDS and non-ARDS patients [7–9]. To our knowledge, no studies have elaborated on individualizing mechanical power according to severity of ARDSs and predicting mortality.

Our primary objective was to assess the impact of individualized MP thresholds on ICU mortality. The secondary objective was to propose a method to individualize mechanical ventilation (MV) settings based on MP and other covariables.

## Methodology

This retrospective study aimed to assess the effects of personalized MP thresholds on ICU mortality among mechanically ventilated patients and to develop an individualized approach for adjusting ventilation settings based on MP and other covariables. Data were sourced from the Amsterdam University Medical Centers Database (AmsterdamUMCdb), which complies with General Data Protection Regulation (GDPR) regulations and contains anonymized patient information. Due to the study's retrospective nature and use of de-identified data, ethical approval was not required by the Institutional Review Board of Amsterdam UMC, and informed consent was waived as no identifiable data were used [10].

### Ethics approval and consent to participate

Amsterdam UMC conducted a comprehensive Data Privacy Impact Analysis of its database in accordance with European and Dutch regulations, including the GDPR. An independent review by experts, including Prof. Sijbrands and the Netherlands Federation of University Medical Centers, confirmed that the database meets GDPR standards with a very low risk of re-identification [10].

An ethics review led by Dr. Erwin Kompanje determined that the use of de-identified medical data is ethically sound, given the minimal risk and adherence to ethical standards in medical research. It was concluded that the benefits of data sharing outweigh the potential risks [10]. As the study involved anonymized data, informed consent was not required, in compliance with national regulations [10].

### Study population

The study encompassed patients aged 18 years or older who needed a minimum of 48 hours of pressure-controlled mechanical ventilation. Exclusions encompassed patients who died or were extubated within the initial 48 hours, as well as those with inadequate data, defined as missing or incomplete key ventilation parameters, essential demographic information, or physiologically implausible values. Based on their PaO2/FiO2 ratio (PFR) over the first six hours of mechanical ventilation, patients were stratified into distinct hypoxemia categories–nonhypoxemic (PFR > 300), mildly hypoxemic (PFR 200–300), moderately hypoxemic (PFR 100–199), and severely hypoxemic (PFR < 100). The study population was analyzed with regard to mortality outcomes.

### Data collection and mechanical power calculation

Baseline patient characteristics, including demographic information, clinical status and laboratory investigations, were meticulously gathered, while mechanical ventilation parameters were also recorded. Data extraction was performed using a structured query language (SQL) query with the amsterdamumcdb Python package, retrieving relevant patient demographics, clinical characteristics, and mechanical ventilation parameters from the Amsterdam UMC database using queries published on the database website.

After extraction, the data underwent a cleaning process to remove entries that did not meet the study inclusion criteria. Erroneous and duplicate entries were identified and excluded based on predefined clinical rules. Missing values were handled using the K-Nearest Neighbors Imputer (KNNImputer) from the scikit-learn library with n_neighbors = 7, which imputes missing data by averaging values from the seven nearest neighbors, preserving the dataset's integrity by leveraging existing patterns.

The time-weighted average (TWA) was calculated over the initial 24 and 48 hours of mechanical ventilation. The mechanical power of ventilation was determined using the surrogate formula [11]:

$$0.098 \times TV \times RR \times PEEP \times \textit{Inspiratory Pressure}$$

The TWA of MP over the first 24 and 48 hours of mechanical ventilation was then normalized to the patient's ideal body weight (IBW) to account for variations in patient size and lung mechanics. This metric provides insight into the potential lung tissue damage associated with mechanical ventilation.

The cleaned dataset was divided into training and validation subsets for various experimental analyses, including mortality prediction models and mechanical power assessments. Normalization of key variables ensured robust and reproducible analyses, standardizing data across diverse patient sizes.

## Statistical analysis and machine learning models

The collected data were described using the mean, standard deviation (SD), median, and interquartile range (IQR) for symmetric and nonsymmetric distributions, respectively. The Mann–Whitney test was employed for nonsymmetric continuous variables. T-tailed test was used for symmetrically distributed variables to compare patients grouped by mortality status, and chi-square was used for categorical data. A P value < 0.05 indicates statistical significance. Unadjusted odds ratios (ORs) were calculated to assess the influence of TWA mechanical power normalized to IBW on mortality.

Safe upper limits for IBW-adjusted MP were determined for each hypoxemia group by applying the Mann-Whitney test. It was applied to compare the distributions of IBW-adjusted MP between mortality groups, focusing on the alternative hypothesis that the variable's distribution was greater among nonsurvivors. Employing a predefined significance level, the test results were evaluated. If the p value was deemed significant, a substantial upper limit value was calculated using the survivors' percentile corresponding to the significance level. These safe upper limits provide a benchmark for clinicians to avoid VILI. By identifying these thresholds, we propose practical guidelines for adjusting ventilation settings to minimize harm while maintaining adequate oxygenation and ventilation. For example, clinicians can use these upper limits to personalize mechanical ventilation, ensuring that mechanical power remains within safe limits for each patient's lung condition. This individualized approach could reduce the risk of VILI and improve patient outcomes by preventing over-ventilation in vulnerable populations, such as those with moderate to severe hypoxemia.

Survival (COX) analysis comparing populations grouped according to the identified limits was performed, including noncollinear variables that had a significant association with mortality. Kaplan–Meier estimator plotted for visualization.

**Mortality prediction.**   First, imputation was performed for data with 10% missing values or less via k-nearest neighbour (k = 7). Afterward, the synthetic minority oversampling technique was used to balance the data between nonsurvivors and survivors [12]. Ventilation-related features are included within the model by default, i.e., MP, IBW-normalized MP, IBW-normalized tidal volume, 48-h standard deviation of MP, PEEP, respiratory rate, and driving pressure. In addition, relevant features are selected via 2 steps: 1) removing features that demonstrated larger than 90% cross-correlation and 2) selecting the 10 most significant features based on random forest on top of the ventilation-related features. Feature selection prioritized clinically relevant variables and those with significant predictive impact, in addition to removing features with high collinearity (>90%). Variables such as lactate levels, age, and IBW-

adjusted MP were chosen based on both their physiological relevance and contribution to model accuracy. Ultimately, various machine learning models were systematically evaluated for their ability to predict ICU mortality, including logistic regression, random forest, Support Vector Machine (SVM), Ada boosting, xgBoost, and stacking [13–16]. No extensive hyper-parameter tuning was performed for the machine learning models. Default parameter settings provided by the respective libraries were used. This approach was chosen to provide an initial proof-of-concept for using machine learning in ICU mortality prediction. Performance evaluation utilized accuracy, precision, recall, and the area under the receiver operating curve (ROC-AUC) to gauge the predictive capabilities of each model.

## Individualization of MV settings

The individualization of the MV setting in the present study aims to trade off over- and under-ventilation by leveraging the mortality prediction model from the previous section. Fig 1 illustrates the optimization scheme and a brief explanation follows below.

A patient is admitted to the ICU, and the responsible physician manually sets the MV settings. The respiratory acidosis status of the patient is then observed intermittently, e.g., by taking measures of end-tidal CO2 or PaCO2, which is then fed into the individualization algorithm. The MV settings are then optimized based on whether the degree of potential acidosis is high or low to maximize minute ventilation (VE) or minimize MP, respectively. The algorithm then provides initial values for tidal volume and/or driving pressure and respiratory rate, which fulfills the desired VE and MP. These values are then optimized further by leveraging the mortality prediction model mentioned in the previous section. This means that the MV settings and other measured covariables are fed iteratively into the mortality prediction until the prediction result is a False, while keeping the desired VE or MP. If this is not possible, then the setting is not changed until some new information regarding the other covariables are updated. In a parallel workflow, automatic PEEP titration is recommended to find the optimal PEEP setting for each patient [17], thereby significantly reducing the degree of freedom within the optimization process. The step-wise minimization of MP ultimately aims to achieve a quicker, data-driven and reproducible ventilator wean.

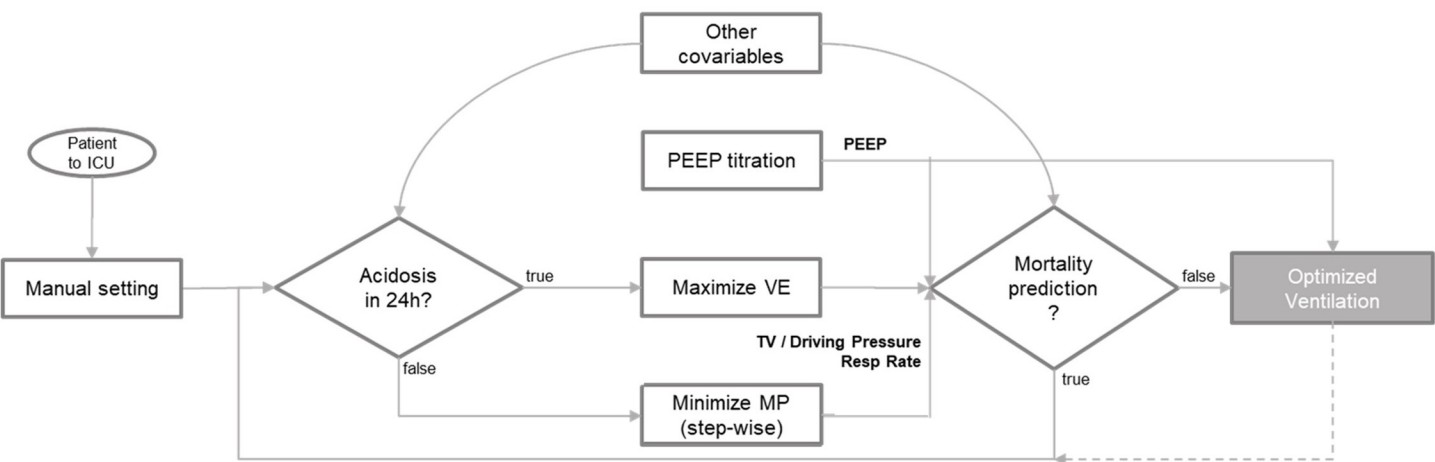

**Fig 1. An overall scheme of the individualization process of mechanical ventilation setting.** ICU: Intensive care unit. PEEP: Positive end-expiratory pressure. VE: Minute ventilation. MP: Mechanical power. TV: Tidal volume.

For example, considering a patient with ARDS and severe acidosis (PaCO2 > 50 mmHg), the algorithm would prioritize increasing minute ventilation by adjusting tidal volume and respiratory rate, while maintaining safe MP limits. In contrast, if PaCO2 is normal but there is a risk of overventilation (e.g., elevated driving pressures), the algorithm would focus on reducing MP to prevent lung injury, while ensuring adequate oxygenation. The algorithm continuously updates settings based on real-time ICU data, such as blood gas levels or respiratory parameters.

## Results

From a total of 23,106 admissions screened from the database, 2,338 admissions met the inclusion and exclusion criteria and were subjected to analysis (Fig 2). These patients were systematically categorized into four distinct hypoxemia groups: nonhypoxemic (615, 26.3%), mildly hypoxemic (736, 31.48%), moderately hypoxemic (840, 35.93%), and severely hypoxemic (147, 6.29%). Detailed baseline characteristics, mechanical ventilation parameters, and laboratory values of the analyzed admissions are presented in Tables 1–3, respectively.

## Mechanical power and mortality

The analysis revealed that nonsurvivors had significantly higher time-weighted average mechanical power (TWA-MP) than survivors ($p < 0.001$), as indicated in Table 3. The 48-hour TWA-MP was significantly associated with ICU mortality with an unadjusted odds ratio (OR) of 1.02 (95% CI: 1.01–1.03) per 1 J/min increase. As detailed in S1 Table, the stepwise multivariate analysis revealed that while the 48-hour TWA-MP had a borderline p-value of 0.098 (95% CI: 0.998–1.024), other covariates such as age and lactate demonstrated much stronger associations with mortality. Specifically, age categories above 60 years and the highest lactate levels

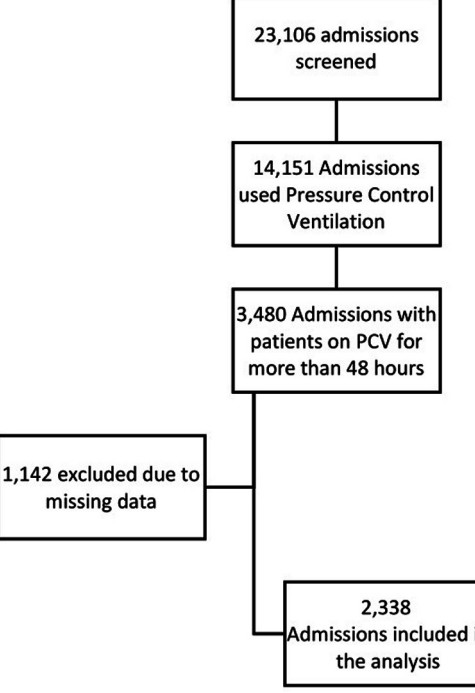

**Fig 2. Patient inclusion flow chart.**

**Table 1.  Characteristics of patients admitted to the ICUBSA, Body Service Area; ICU, Intensive Care Unit.**

| Variable | Dead | Alive | P Value |
|---|---|---|---|
| Gender, count (%) | | | 0.380 |
| Male | 450 (65.4%) | 1112 (67.4%) | |
| Female | 238 (34.6%) | 538 (32.6%) | |
| Age in years, count (%) | | | 0.000 |
| 18–39 | 44 (6.4%) | 248 (15.03%) | |
| 40–49 | 44 (6.4%) | 207 (12.55%) | |
| 50–59 | 103 (14.97%) | 312 (18.91%) | |
| 60–69 | 164 (23.84%) | 392 (23.76%) | |
| 70–79 | 217 (31.54%) | 346 (20.97%) | |
| > 80 | 116 (16.86%) | 145 (8.79%) | |
| Height (Cm), median (IQR) | 175.00 (165.00–185.00) | 175.00 (165.00–185.00) | 0.016 |
| Weight (kg), median (IQR) | 75.00 (65.00–85.00) | 85.00 (75.00–85.00) | 0.000 |
| BSA, median (IQR) | 1.91 (1.78–2.09) | 1.97 (1.85–2.09) | 0.000 |
| Ideal body weight, median (IQR) | 70.57 (61.47–79.67) | 70.57 (61.47–79.67) | 0.023 |
| SOFA respiratory score, mean (SD) | 1.94 (0.29) | 1.89 (0.38) | 0.002 |
| SOFA total score, mean (SD) | 7.94 (2.65) | 7.07 (2.50) | 0.000 |
| Urgency of ICU admission, count (%) | | | 0.284 |
| Not urgent | 295 (42.9%) | 749 (45.4%) | |
| Urgent | 393 (57.1%) | 901 (54.6%) | |
| Hypoxemia status, count (%) | | | 0.038 |
| None hypoxic | 156 (22.7%) | 459 (27.8%) | |
| Mild hypoxemia | 219 (31.8%) | 517 (31.3%) | |
| Moderate hypoxemia | 272 (39.5%) | 568 (34.4%) | |
| Sever hypoxemia | 41 (6%) | 106 (6.4%) | |
| Admitting speciality, count (%) | | | 0.000 |
| Cardio Surgery | 52 (7.56.7%%) | 18.7%4 (11.15%) | |
| Cardiology | 208 (30.23%) | 279 (16.91%) | |
| General Surgery Gastroenterology | 35 (5.09%) | 108 (6.55%) | |
| General Surgery Lungs | 8 (1.16%) | 24 (1.45%) | |
| Hematology | 28 (4.07%) | 27 (1.64%) | |
| Adult Intensive Care | 90 (13.08%) | 213 (12.91%) | |
| Internal Medicine | 54 (7.85%) | 119 (7.21%) | |
| Otorhinolaryngology | 11 (1.6%) | 52 (3.15%) | |
| Pulmonology | 20 (2.91%) | 54 (3.27%) | |
| Neurosurgery | 51 (7.41%) | 225 (13.64%) | |
| Neurology | 26 (3.78%) | 59 (3.58%) | |
| Trauma | 24 (3.49%) | 157 (9.52%) | |
| Vascular Surgery | 46 (6.69%) | 75 (4.55%) | |
| Others | 35 (5.09%) | 74 (4.48%) | |
| ICU admission count, mean (SD) | 1.00 (0.07) | 1.01 (0.09) | 0.233 |
| Length of stay (days), median (IQR) | 178.00 (92.00–382.25) | 334.00 (199.25–555.00) | 0.000 |
| Mechanical ventilation days, median (IQR) | 122.07 (74.00–278.20) | 157.14 (87.00–310.00) | 0.001 |

significantly influenced mortality, with older age groups (e.g., >80 years) exhibiting an odds ratio (OR) of 5.86 (p<0.001). The multivariate model suggests that although elevated MP is a contributor, its effect is overshadowed by systemic markers of disease severity, such as lactate (OR: 1.165, p<0.001). These findings underscore the multifactorial nature of ICU mortality,

**Table 2. Laboratory values of the study population during admission.**

| Variable | Dead Median (IQR) | Alive Median (IQR) | P Value |
|---|---|---|---|
| Highest WBC | 17.65 (13.10–22.92) | 16.60 (12.90–21.80) | 0.0581 |
| Lowest WBC | 8.50 (5.57–11.70) | 8.40 (6.10–11.43) | 0.3892 |
| Highest Hemoglobin | 8.20 (7.40–9.10) | 8.20 (7.40–9.10) | 0.6139 |
| Lowest Hemoglobin | 5.65 (4.90–6.80) | 5.60 (4.90–6.70) | 0.7235 |
| Highest Thrombocytes | 239.50 (178.00–310.25) | 237.00 (183.00–306.00) | 0.4168 |
| Lowest Thrombocytes | 128.50 (73.00–193.25) | 135.00 (85.00–196.00) | 0.0256 |
| Highest ASAT | 145.00 (55.00–411.75) | 70.00 (35.00–198.00) | 0.0000 |
| Highest Bilirubin | 12.00 (8.00–22.50) | 12.00 (8.00–19.00) | 0.1624 |
| Highest APTT | 74.00 (48.00–152.00) | 53.00 (43.00–102.00) | 0.0000 |
| Lowest APTT | 38.00 (33.00–44.00) | 35.00 (32.00–41.00) | 0.0000 |
| Highest Prothrombin time | 1.71 (1.39–2.47) | 1.54 (1.29–2.01) | 0.0000 |
| Highest CRP | 103.00 (23.00–216.00) | 96.00 (14.50–215.00) | 0.2054 |
| Highest Lactate | 5.90 (2.70–9.50) | 3.20 (1.70–6.05) | 0.0000 |
| Highest Troponin | 0.24 (0.05–1.56) | 0.17 (0.03–1.00) | 0.0001 |
| Highest Creatinine | 142.00 (96.00–226.25) | 107.00 (81.00–165.00) | 0.0000 |
| Highest Urea | 11.05 (7.43–16.27) | 8.10 (5.60–12.53) | 0.0000 |
| Highest Potassium | 4.90 (4.60–5.30) | 4.70 (4.40–5.20) | 0.0000 |
| Lowest Potassium | 3.50 (3.40–3.70) | 3.50 (3.40–3.70) | 0.8135 |
| Highest Sodium | 145.00 (142.00–149.00) | 145.00 (142.00–148.00) | 0.5128 |
| Lowest Sodium | 135.50 (132.00–138.00) | 136.00 (133.00–138.00) | 0.3562 |
| Highest pH | 7.44 (7.40–7.48) | 7.45 (7.42–7.48) | 0.0001 |
| Lowest pH | 7.17 (7.17–7.25) | 7.23 (7.17–7.29) | 0.0000 |
| Highest Pco2 | 56.00 (49.00–61.00) | 53.00 (48.00–61.00) | 0.0000 |
| Lowest Pco2 | 29.00 (29.00–33.00) | 31.00 (29.00–35.00) | 0.0000 |
| Highest BE | 9.30 (4.00–11.70) | 6.20 (3.20–10.20) | 0.0000 |
| Lowest BE | 0.10 (-7.40–1.90) | 0.00 (-6.10–0.50) | 0.0046 |

WBC: White blood cell. ASAT: Aspartate aminotransferase. APTT: Activated partial thromboplastin clotting time. CRP: C-reactive protein. BE; Base Excess.

where MP alone cannot act as an independent predictor due to the interplay of patient-specific factors. These findings suggest that while elevated mechanical power can contribute to patient outcomes, it may not act as an independent predictor of mortality in this cohort. Subsequent analysis based on hypoxemia status demonstrated that TWA-MP for the initial 48 hours was significantly associated with mortality across all groups except for patients with severe hypoxemia, as displayed in Table 4.

## Safe upper limits of IBW-adjusted MP

A safe MP upper limit of 16.51 J/min was calculated for all patients. Safe upper limits of IBW-adjusted MP (J/min/kg) were established for different hypoxemia groups: 0.22 (nonhypoxemic), 0.27 (mild hypoxemia), and 0.34 (moderate hypoxemia). Notably, a statistically significant upper limit could not be identified for severely hypoxemic patients.

## COX-survival analysis

Cox survival analysis utilizing the upper limits of IBW-adjusted MP showed no significant influence of 48 h-TWA-MP on mortality (p = 0.35). The hazard ratio (HR) calculated was 0.52

**Table 3. Mechanical ventilation measurements in the first 24 and 48 hours of admission.**

| Variable | Dead, median (IQR) | Alive, median (IQR) | P Value |
|---|---|---|---|
| **Mechanical ventilation measurements in the initial 24 hours of IMV** | | | |
| TWA PFR | 213.83 (167.16, 276.45) | 230.58 (180.87, 293.99) | 0.000 |
| TWA respiratory rate (breath/min) | 19.82 (16.53, 23.15) | 18.51 (15.89, 21.95) | 0.000 |
| TWA inspiratory time (seconds) | 1.07 (0.90, 1.30) | 1.12 (0.95, 1.31) | 0.002 |
| TWA IE ratio | 0.52 (0.50, 0.57) | 0.51 (0.49, 0.56) | 0.010 |
| TWA PEEP | 9.08 (6.49, 11.78) | 8.18 (5.57, 11.45) | 0.001 |
| TWA tidal volume (liters) | 0.46 (0.41, 0.51) | 0.48 (0.43, 0.54) | 0.000 |
| TWA tidal volume (ml/kg IBW) | 6.62 (5.94, 7.57) | 6.85 (6.08, 7.78) | 0.004 |
| TWA driving pressure | 14.59 (11.54, 18.10) | 13.97 (11.35, 16.87) | 0.002 |
| TWA peak pressure | 23.69 (18.98, 29.51) | 22.30 (18.05, 27.80) | 0.000 |
| TWA mechanical power (J/min) | 20.87 (15.15, 29.10) | 19.27 (14.27, 26.95) | 0.004 |
| TWA mechanical power for IBW (J/min/kg) | 0.31 (0.22, 0.42) | 0.28 (0.20, 0.39) | 0.001 |
| **Mechanical ventilation measurements in the initial 48 hours of IMV** | | | |
| TWA PFR | 215.42 (166.67, 271.39) | 228.68 (183.76, 283.71) | 0.000 |
| TWA respiratory rate (breath/min) | 19.74 (17.20, 23.54) | 18.52 (16.04, 22.08) | 0.000 |
| TWA inspiratory time (seconds) | 1.06 (0.90, 1.25) | 1.11 (0.94, 1.29) | 0.000 |
| TWA IE ratio | 0.52 (0.50, 0.58) | 0.51 (0.49, 0.56) | 0.001 |
| TWA PEEP | 9.36 (6.92, 11.90) | 8.62 (6.12, 11.32) | 0.001 |
| TWA tidal volume (liters) | 0.47 (0.42, 0.52) | 0.49 (0.44, 0.54) | 0.000 |
| TWA tidal volume (ml/kg IBW) | 6.69 (5.96, 7.68) | 6.93 (6.16, 7.85) | 0.001 |
| TWA driving pressure | 14.39 (11.77, 17.85) | 13.45 (11.00, 16.44) | 0.000 |
| TWA peak pressure | 23.94 (19.28, 29.39) | 22.16 (18.09, 27.43) | 0.000 |
| TWA mechanical power (J/min) | 21.39 (16.34, 29.58) | 19.54 (14.64, 26.68) | 0.000 |
| TWA mechanical power for IBW (J/min/kg) | 0.31 (0.24, 0.43) | 0.28 (0.21, 0.39) | 0.000 |

IMV: Invasive mechanical ventilation. TWA: Time-weighted average. PFR: PaO2/FiO2 Ratio. IE ratio: Inspiratory-expiratory ratio. PEEP: Positive end-expiratory pressure. IBW: Ideal body weight.

(0.13–2.06, 95% CI). This implies that for each unit increase in IBW-adjusted MP, the hazard of mortality decreases by 48%. This analysis suggested that mechanical power within the established safe limits did not substantially impact survival. The non-significance may be attributed to the limited sample size in specific subgroups, such as severely hypoxemic patients, where mortality is driven by underlying pathology rather than MP. According to the determined safe limits of 48 h-TWA and IBW-adjusted MP, the COX-survival curve revealed differences in mortality at 90 days of hospital stay. Patients with no or mild hypoxia that received lower MP had better survival at 90 days. On the other hand, among severely hypoxic patients, those who received mechanical power below the identified safe limit exhibited higher and earlier mortality, as depicted in Fig 3.

## Machine learning models and mortality prediction

Utilizing 5-fold cross-validation, Table 5 compiles the predictive performance of the models tested. Although the Stacking method demonstrated marginally better accuracy (0.80 vs. 0.78 for XGBoost), XGBoost was chosen due to its computational efficiency and ease of implementation in real-time settings. The slight performance trade-off was considered acceptable given the practical benefits of faster training and prediction times, critical for ICU applications. Thus, the XGBoost model is considered the best choice to be used in the subsequent

**Table 4. Mechanical power according to hypoxemia status and mortality.**

| Group | Variable | Dead, median (IQR) | Alive, median (IQR) | P Value |
|---|---|---|---|---|
| **Mechanical power for the initial 24 hours of IMV** | | | | |
| Nonhypoxic | TWA-MP | 15.79 (12.33–19.77) | 14.82 (11.82–19.4) | 0.201 |
| | TWA-MP/IBW | 0.22 (0.17–0.31) | 0.21 (0.17–0.28) | 0.097 |
| Mild Hypoxemia | TWA-MP | 18.27 (14.52–25.71) | 18.11 (14.03–24.72) | 0.22 |
| | TWA-MP/IBW | 0.29 (0.21–0.39) | 0.26 (0.2–0.37) | 0.069 |
| Moderate Hypoxemia | TWA-MP | 24.42 (19.08–32.68) | 23.21 (17.7–31.55) | 0.196 |
| | TWA-MP/IBW | 0.36 (0.27–0.47) | 0.34 (0.25–0.46) | 0.099 |
| Sever Hypoxemia | TWA-MP | 31.62 (21.93–41.71) | 33.99 (25.13–41.15) | 0.743 |
| | TWA-MP/IBW | 0.48 (0.32–0.58) | 0.49 (0.38–0.6) | 0.572 |
| **Mechanical power for the initial 48 hours of IMV** | | | | |
| Nonhypoxic | TWA-MP | 16.97 (13.17–21.75) | 15.37 (12.01–19.66) | 0.005 |
| | TWA-MP/IBW | 0.24 (0.18–0.33) | 0.22 (0.17–0.29) | 0.004 |
| Mild Hypoxemia | TWA-MP | 19.81 (15.49–26.19) | 18.96 (14.44–24.37) | 0.078 |
| | TWA-MP/IBW | 0.29 (0.23–0.39) | 0.27 (0.21–0.36) | 0.023 |
| Moderate Hypoxemia | TWA-MP | 25.45 (18.85–33.01) | 23.77 (17.87–30.77) | 0.012 |
| | TWA-MP/IBW | 0.36 (0.29–0.47) | 0.34 (0.25–0.44) | 0.005 |
| Sever Hypoxemia | TWA-MP | 30.31 (21.14–43.95) | 29.72 (24.36–39.21) | 0.607 |
| | TWA-MP/IBW | 0.44 (0.32–0.61) | 0.43 (0.35–0.55) | 0.870 |

IMV: Invasive mechanical ventilation. TWA-MP: Time-weighted average mechanical power. IBW: Ideal body weight.

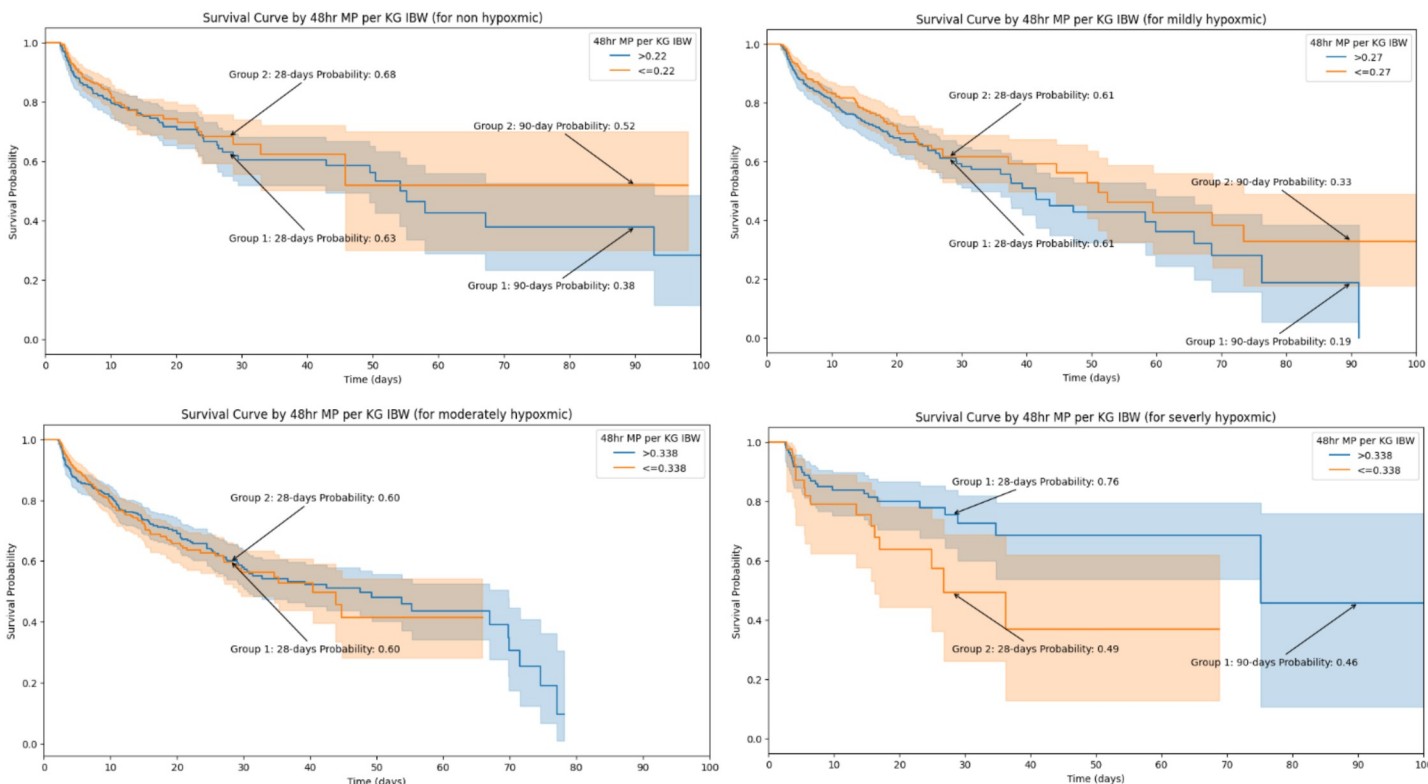

**Fig 3. COX-survival curve according to the found safe limits of 48 h-TWA, IBW-adjusted MP.** Stratified into hypoxemic levels: nonhypoxemic (top left), mildly hypoxemic (top right), moderately hypoxemic (bottom left), and severely hypoxemic (bottom right). TWA: Time-weighted average. IBW: Ideal body weight. MP: Mechanical power.

**Table 5. Prediction performance of each model.**

| Model tested | Accuracy | Precision | Recall | ROC AUC |
|---|---|---|---|---|
| Logistic Regression | 0.67 | 0.67 | 0.67 | 0.74 |
| Random Forest | 0.79 | 0.79 | 0.79 | 0.86 |
| SVM | 0.62 | 0.61 | 0.67 | 0.68 |
| AdaBoosting | 0.74 | 0.74 | 0.73 | 0.82 |
| XGBoost | 0.78 | 0.79 | 0.76 | 0.88 |
| Stacking | 0.80 | 0.80 | 0.79 | 0.87 |

SVM: Support Vector Machine.

mechanical ventilation individualization study among the prediction models (accuracy: 0.78, precision: 0.79, recall: 0.76, AUROC: 0.88). Additionally, the SHapley Additive exPlanations (SHAP) values were employed to assess the sensitivity of the XGBoost model's predictions to selected covariables [18]. This analysis suggested that the adjusted 48 h-TWA-MP had a relatively limited effect compared to other covariables, as demonstrated in Fig 4.

### Individualization method case study

The proposed method for individualizing MV settings was applied to nonsurvivors. In this case study, the acidosis status is defined by a $PaCO2$ larger than 45 mmHg and a blood pH lower than 7.35. This case study demonstrated a reduction in predicted mortality, with 58 more survivors predicted out of the initially identified 614 nonsurvivors (9.4%).

## Discussion

This study investigated the impact of personalized MP thresholds on ICU mortality in mechanically ventilated patients. These findings, along with other data, were used to help develop a machine learning model, which is part of a broader, newly suggested individualized approach to optimize ventilation settings and improve survival outcomes. The primary findings of this study include a significant association between IBW-adjusted MP and ICU mortality, the identification of safe upper limits for IBW-adjusted MP in all hypoxemic groups except those with severe hypoxemia, and the high accuracy of XGBoost in predicting ICU mortality. Despite the association between MP and mortality, MP played a relatively limited role compared to other covariables in the prediction model. In a case study, when the newly suggested individualized approach was applied to nonsurvivors in the current dataset, a 9.4% reduction in predicted mortality was observed. This suggests a potential survival benefit with the adoption of this approach, bringing the findings closer to bedside clinical application.

### Mechanical power and mortality

The analysis revealed that nonsurvivors had significantly higher 48-hour TWA-MP than survivors, supporting the existing literature that links elevated mechanical power to VILI and increased mortality risk [9,19,20]. Gattinoni et al. emphasized the importance of normalizing MP to lung size for clinical relevance, prompting us to adjust 48h TWA-MP to IBW, which showed an odds ratio of 1.02 per 1 J/min increase, further underscoring the incremental mortality risk associated with higher mechanical power. However, when analyzing different hypoxemic groups, the association between MP and mortality was confirmed in all but the severely hypoxemic group. The lack of association between MP and mortality could be attributed to several factors. These patients are often subjected to more aggressive lung-protective

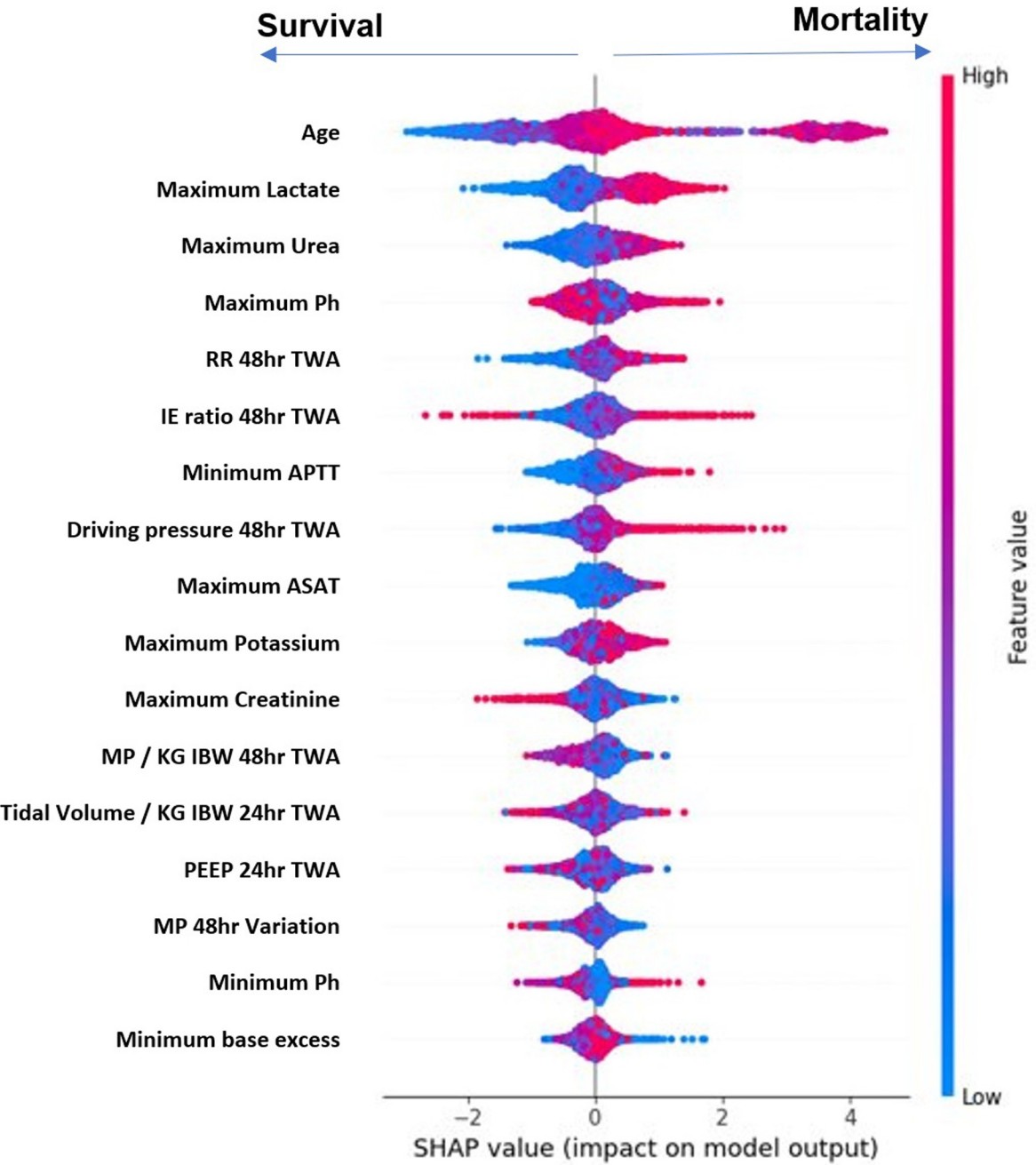

**Fig 4. SHAP values display the sensitivity of the XGBoost model in predicting ICU mortality against different covariables.** TWA: Time Weighted Average. APPT: Activated Partial Thromboplastin Clotting Time. ASAT: Aspartate Aminotransferase. MP: Mechanical power. IBW: Ideal Body Weight. PEEP: Positive End Expiratory Pressure.

ventilation strategies, including lower tidal volumes and higher PEEP, aimed at minimizing further lung damage. This might reduce the relative impact of mechanical power on their overall mortality. Additionally, the underlying lung pathology in these patients, such as extensive consolidation or fibrosis, could make the lungs less responsive to mechanical power variations. Moreover, the higher mortality in this group might be driven more by the severity of their

underlying disease and refractory hypoxemia, rather than the mechanical ventilation parameters themselves. This is consistent with some previous studies, such as one by Coppola et al., which also failed to demonstrate this association, except after normalization to CT-measured well-inflated lung tissue and respiratory system compliance [4].

### Thresholds for harm in IBW-adjusted MP

We established safe upper limits for IBW-adjusted MP (J/min/kg), which varied according to the patient's hypoxemia status: 0.22 for non-hypoxemic, 0.27 for mildly hypoxemic, and 0.34 for moderately hypoxemic groups. For an average IBW of 70.57 kg (as observed in the dataset), the corresponding MP thresholds (J/min) are 15.53, 19.05, and 23.99, respectively. These thresholds are consistent with previous findings; for example, Neto et al. identified that MP values exceeding 17 J/min are associated with increased mortality in critically ill patients undergoing invasive ventilation for at least 48 hours [7]. However, this study advances the field by stratifying these MP thresholds according to hypoxemia severity, offering more specific guidance for clinicians to adjust ventilation settings based on the patient's respiratory condition. While previous studies provided generalized MP values, the individualized approach in this study highlights the importance of tailoring mechanical ventilation strategies to the specific characteristics of the patient, such as lung injury severity.

Furthermore, a statistically significant upper limit for severely hypoxemic patients was not determined due to the lack of a significant association between MP and mortality within this group. This finding suggests that in severely hypoxemic patients, factors beyond MP, such as underlying lung pathology, disease severity, and the use of lung-protective strategies, may play a more dominant role in determining outcomes. This study underscores the complexity of MP's impact in this subgroup, suggesting that mechanical power alone may not be the most relevant predictor of mortality in these patients.

### Machine learning models and mortality prediction

Unlike commonly used ICU prediction models that are typically based on data collected within 24 hours pre or post-ICU admission, ML models provide higher flexibility, and adapt and update predictions based on how patients respond to treatments after admission. Among the machine learning models, XGBoost demonstrated robust performance in predicting ICU mortality. This is consistent with other ML ICU mortality prediction models, irrespective of the origin of data, database used or age group, possibly because XGBoost is a better classifier for imbalanced datasets. The inclusion of ventilation-related features and other significant covariables in the model likely contributed to its predictive power. 48 h-TWA-MP had a relatively limited effect compared to other covariables, suggesting that ICU mortality is a multifactorial issue that mechanical power may not be able to solely predict or influence.

### Individualization method case study

The individualized approach to mechanical ventilation showed promise in reducing predicted mortality. Despite the results that show MP might only have a relatively limited effect on mortality, there is a statistically significant difference in MP between survivors and nonsurvivors. Thus, the proposed individualization approach that optimizes for reducing MP is still clinically relevant to reduce mortality.

By utilizing dynamic optimization based on multiple physiological variables to optimize MV in real-time, we could achieve a more nuanced and patient-specific ventilatory strategy, potentially improving outcomes.

## Clinical implications and future directions

Our findings are a step towards the realization of personalized mechanical ventilation strategies that consider individualized MP thresholds as part of routine clinical practice. Implementing real-time dynamic optimization of ventilatory settings could minimize VILI and potentially improve survival outcomes in critically ill patients. However, there are several challenges to consider when adopting this individualized approach in practice. First, integrating machine learning algorithms into clinical workflows requires robust computational infrastructure and real-time data collection, which may not be feasible in all healthcare settings. Second, clinicians will need training to interpret and act on the recommendations provided by these models, particularly as these strategies differ from traditional ventilation protocols. Additionally, patient variability and rapidly changing clinical conditions could make it difficult to standardize such an approach across diverse ICU environments.

Future research should focus on validating these MP thresholds in diverse patient populations and clinical settings to establish their generalizability and clinical utility. Prospective studies are needed to test the efficacy of machine learning-driven individualized approaches in clinical practice. Randomized controlled trials could evaluate the impact of personalized MP thresholds on patient outcomes compared to standard care. Additionally, integrating continuous monitoring and adjustment of MP in a feedback loop, along with other physiological variables, may help refine ventilatory strategies further, reducing mortality and enhancing patient care.

## Limitations

This study has several limitations, including its retrospective, observational design and the use of the single-center Amsterdam UMC database. The Amsterdam UMC database is a relatively new critical care database that contains approximately 1 billion data points from over 20,000 patients admitted between 2003 and 2016. The database's single-center nature, lack of supporting medical history, and relatively small size limit the generalizability of the findings. Additionally, the cohort is heavily weighted toward patients who underwent cardiothoracic surgery, which may not reflect the broader ICU population. The time period from which the data were collected also introduces some heterogeneity in treatment protocols and patient cohorts. The lack of external validation further limits the generalizability of our results. Without testing the predictive models and mechanical power thresholds on external or multi-center datasets, it is unclear whether these findings would apply to other populations with different clinical characteristics or ventilation practices. Future studies should aim to validate these findings using external datasets to ensure broader applicability and to strengthen the robustness of the proposed individualized ventilation strategies. Additionally, our exclusion of patients who died or were extubated within the first 48 hours and the focus solely on pressure-controlled ventilation may introduce selection bias. The surrogate formula used for calculating mechanical power also has inherent limitations and assumptions. Validation with external databases and incorporating other variables, such as radiological features, could improve both the generalizability and precision of our findings.

## Conclusions

Our study provides valuable insights into the relationship between mechanical power and ICU mortality. Identifying safe upper limits for IBW-adjusted MP and the promising results from machine learning models in predicting mortality underscore the potential for more personalized and data-driven approaches in managing mechanically ventilated patients. However, the project was limited by the singular dataset used, and further validation against other ICU datasets and multi-center clinical validation is required.

## Supporting information

**S1 Table. Multivariate stepwise backward analysis.**
(DOCX)

## Acknowledgments

The authors would like to thank the Amsterdam University Medical Centers Database (AmsterdamUMCdb) staff and contributors for their support in data collection and analysis.

## Author Contributions

**Conceptualization:** Ahmed S. Alkhalifah, Kenny Rumindo, Edgar Brincat, Florian Blanchard, David Clarke, Benjamin Popoff, Olivier Duranteau, Zubair Umer Mohamed, Abdelrahman Senosy.

**Data curation:** Ahmed S. Alkhalifah, Kenny Rumindo, Abdelrahman Senosy.

**Formal analysis:** Ahmed S. Alkhalifah, Kenny Rumindo.

**Methodology:** Kenny Rumindo, Edgar Brincat, Florian Blanchard, Johan Helleberg, David Clarke, Benjamin Popoff, Olivier Duranteau, Zubair Umer Mohamed, Abdelrahman Senosy.

**Project administration:** Abdelrahman Senosy.

**Software:** Kenny Rumindo.

**Supervision:** Ahmed S. Alkhalifah.

**Visualization:** Ahmed S. Alkhalifah.

**Writing – original draft:** Ahmed S. Alkhalifah, Edgar Brincat, Florian Blanchard, Johan Helleberg, David Clarke, Benjamin Popoff, Olivier Duranteau, Zubair Umer Mohamed, Abdelrahman Senosy.

**Writing – review & editing:** Ahmed S. Alkhalifah, Edgar Brincat, Florian Blanchard, Johan Helleberg, David Clarke, Benjamin Popoff, Olivier Duranteau, Zubair Umer Mohamed, Abdelrahman Senosy.

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
