## [Decision Letter · Decision Letter 0]

30 Jul 2024

PONE-D-24-20623Optimizing mechanical ventilation: personalizing mechanical power to reduce icu mortality - a retrospective cohort studyPLOS ONE

Dear Dr. alkhalifah,

Thank you for submitting your manuscript to PLOS ONE. After careful consideration, we feel that it has merit but does not fully meet PLOS ONE’s publication criteria as it currently stands. Therefore, we invite you to submit a revised version of the manuscript that addresses the points raised during the review process.

**ACADEMIC EDITOR:**

lease carefully assess all the reviewers comments

We look forward to receiving your revised manuscript.

Kind regards,

Silvia Fiorelli

Academic Editor

PLOS ONE

Journal Requirements:

2. You indicated that ethical approval was not necessary for your study. We understand that the framework for ethical oversight requirements for studies of this type may differ depending on the setting and we would appreciate some further clarification regarding your research. Could you please provide further details on why your study is exempt from the need for approval and confirmation from your institutional review board or research ethics committee (e.g., in the form of a letter or email correspondence) that ethics review was not necessary for this study? Please include a copy of the correspondence as an ""Other"" file.

3. In the online submission form, you indicated that [The datasets generated and analyzed for this study are available from the corresponding author on request.]. 

Reviewers' comments:

Reviewer's Responses to Questions

**Comments to the Author**

1. Is the manuscript technically sound, and do the data support the conclusions?

Reviewer #1: Yes

Reviewer #2: Partly

Reviewer #3: Partly

2. Has the statistical analysis been performed appropriately and rigorously? 

Reviewer #1: Yes

Reviewer #2: No

Reviewer #3: Yes

3. Have the authors made all data underlying the findings in their manuscript fully available?

Reviewer #1: Yes

Reviewer #2: Yes

Reviewer #3: No

4. Is the manuscript presented in an intelligible fashion and written in standard English?

Reviewer #1: Yes

Reviewer #2: Yes

Reviewer #3: Yes

5. Review Comments to the Author

Reviewer #1: It is a good study about personalising mechanical power in hypoxic patients with use of advanced statistical and machine learning methods. It has also focused on individualised patient care, aligning with modern precision medicine trends. I agree with the limitations you have highlighted.

Reviewer #2: Thank you for the interesting and potentially impactful study. There are some concerns about this study.

1. Please provide more details on the machine learning models, including feature selection process and hyperparameter tuning.

2. Please expand on the proposed individualization method for mechanical ventilation settings. How exactly would this be implemented clinically?

3. The authors should include a multivariable analysis to assess the independent association of mechanical power with mortality after adjusting for confounders.

4. Please consider external validation of the findings using another ICU dataset, if possible.

5. The authors should clarify how the "safe upper limits" for mechanical power were determined statistically.

6. The counter-intuitive findings in severely hypoxemic patients warrant further discussion and analysis.

7. More context on how these findings compare to existing literature on mechanical power thresholds would be needed.

8. The clinical implications and potential next steps for research should be elaborated on in the discussion.

Reviewer #3: The manuscript investigates the relationship between the Mechanical Power (MP) during mechanical ventilation and ICU mortality. The authors analyzed data from 2338 patients in the Amsterdam University Medical Centers Database and employed statistical and machine learning techniques. The authors found that nonsurvivors had significantly higher time-weighted average mechanical power (TWA-MP) than survivors, defined safe upper limits for IBW-adjusted MP for different hypoxemia groups and developed an individualized approach to change mechanical ventilation settings to reduce predicted mortality.

While the manuscript presents valuable insights, several areas could benefit from further refinement to enhance clarity, reproducibility, and impact. The strengths of the study lie on the large sample size from a critical care database, the good statistical methodology to define the cohort and the proposed novel approach to individualize mechanical ventilation settings. However, the manuscript presents several weaknesses: the primary outcome is ICU mortality, and this might not be the most appropriate measure for studying the effects of the first 48h of Mechanical Power; the methodology is not detailed and lacks basic data pipeline explanations; the results of the survival analysis with safe upper limits for IBW-adjusted MP on different hypoxemia groups show no statistically significance, what rejects one conclusion of the paper; the background and discussion of the manuscript are not explored in depth; and finally, it appears that there is no connection between the identification of the upper MP limit and the individualization of mechanical ventilation settings in the methodology of the manuscript.

Taking all of this into account, my recommendation for the manuscript is: Major Revision.

I wish that these comments find the authors of the paper well, I found very interesting their approach to individualize the MV settings. In order to help the authors to improve the manuscript, the authors should deal with or clarify the following issues.

Major issues:

1. Clarity in Background and Literature Context: The background section is concise and informative on the research but lacks depth in contextualizing existing literature. It gives the impression that the field of mechanical ventilation (including MP) and it’s relation to ICU mortality is less explored by machine learning and deep learning that it actually is. I would recommend expanding the background to include a more comprehensive review.

2. Wrong primary outcome: The research uses the ICU mortality as primary outcome, and this might not be the best outcome for this type of study. ICU mortality relies on several factors not related to mechanical ventilation, and patients with long stays at the ICU usually not die because of bad mechanical ventilation. Usually, other outcomes are used in this type of studies because the represent better a bad ventilation, like ventilator free days, time to successful extubation, or short or medium term mortality.

3. Data transparency: The journal inquires in the guidelines about the importance on data sharing, while it is understandable that the data does not belong to the researchers, it would be ideal that the researchers share the code to extract the data from the original database.

4. Methodological transparency. The methodology of the manuscript lacks detail in data processing, cleaning, missing filling and division in datasets. All of this information is needed to be able to reproduce the research. The subsection “Mortality prediction” of the manuscript it’s the only section in which a bit of the methodology of the data pipeline is explained and it clearly lacks detail. My recommendation would be to write one subsection dedicated to the data pipeline, and separate it from the methodology of the different data experiments.

5. Conclusion not supported by the results. The results in the survival analysis and figure 3 show that there is NO statical significance on the survival probability based on the upper limits identified. The researchers are mistaken and the conclusion based on this result is wrong.

Minor issues:

1. Definition of inadequate data. In abstract and methodology (line number 87) the authors write that they eliminated patients with inadequate data. The manuscript should be more concrete on what is inadequate data and how the authors defined it and processed it.

2. Data extraction. The authors talk about data extraction only at line 93, but code is unavailable and there is almost not real description on how data was extracted (procedure, technical details, tools used). To be able to reproduce the research, this is needed.

3. Statistical and Machine Learning analysis. The authors state at the lines 124-126 that they developed several models. Sadly, they only expose the results of XGBoost. Results of all models should be shown in results to contextualize the good results from XGBoost.

4. Relation between identified MP limits and individualization methods. While the individualization methods for mechanical ventilation settings are a very interesting approach, the authors don’t seem to connect the identified MP limits with the individualization methods. It would be appropriate if they could relate both parts of the paper, if not, is like two different investigations separated and they might want to focus on one for the paper.

5. Result of general limit for MP is not explained on the methodology. At line 167, the authors state that an upper limit of 16.51 J/min was determined for all patients, nonetheless, the authors don’t mention on the methodology how they determine the limit.

6. Provide more details on the algorithm or function used for optimizing MV settings based on MP and other physiological variables. In the subsection of the methodology “Individualization of MV settings”, the authors explain the process of individualization of MV settings, but they don’t explicitly say how the new values of MV are calculated. Ar they random approximation? A inequation being solved? Suggested by any algorithm? They should explain more this part.

7. More in-depth discussion. During the discussion, the authors mainly repeat the results obtained. While I understand that they want to reinstate their results, they should focus more on the clinical meaning of this results.

8. More in-depth limitations. More limitations of this study should be mentioned like that the research is done only on PCV patients, the use ICU mortality outcome (or not if changed to sorth/medium days ICU mortality), and potential biases in the individualization method’s predictions.

9. Table 1. This table has feature “Admitting speciality” with over 10 categories. Some of these categories have less than 10 patients between both groups. As a data cleaning process this feature should be grouped into main categories and “other” category.

10. Table 2. Table has laboratory data, but the extraction is not mentioned in methodology. It should be added to the methodology if this is important.

Conclusion:

The manuscript presents valuable insights into the personalization of mechanical ventilation settings to reduce ICU mortality. However, significant revisions are needed to improve clarity, methodological transparency, and alignment between results and conclusions. Addressing these points will enhance the manuscript's quality and impact.

6. PLOS authors have the option to publish the peer review history of their article (what does this mean?). If published, this will include your full peer review and any attached files.

Reviewer #1: No

Reviewer #2: No

Reviewer #3: **Yes: **Manuel Ruiz-Botella

---

## [Author Response · Author response to Decision Letter 0]

14 Sep 2024

Response to Reviewers

We would like to express our sincere gratitude to the reviewers for their valuable feedback and insightful comments on our manuscript. Your detailed review and constructive suggestions have been instrumental in improving the quality of our work. We have carefully considered each comment and have made the necessary revisions as outlined in our detailed responses below.

Reviewer #1: had no comments

Reviewer #2:

1. Please provide more details on the machine learning models, including feature selection process and hyperparameter tuning.

Details about the model has been added (Line 137-148).

2. Please expand on the proposed individualization method for mechanical ventilation settings. How exactly would this be implemented clinically?

More detailed explanation was added (Line 150-166)

3. The authors should include a multivariable analysis to assess the independent association of mechanical power with mortality after adjusting for confounders.

Added as a supplementary table (S1 Table)

4. Please consider external validation of the findings using another ICU dataset, if possible.

Thank you for your suggestion. We agree that external validation using another ICU dataset would enhance the robustness of our findings. However, due to current data access constraints, conducting external validation is not feasible within the scope of this study. We have thoroughly validated our findings internally and have highlighted the need for external validation in future research as a limitation in the revised manuscript.

5. The authors should clarify how the "safe upper limits" for mechanical power were determined statistically.

Details added at line 127-132

6. The counter-intuitive findings in severely hypoxemic patients warrant further discussion and analysis.

Details added at line 253-255

7. More context on how these findings compare to existing literature on mechanical power thresholds would be needed.

Details added at line 262-264

8. The clinical implications and potential next steps for research should be elaborated on in the discussion.

Details added at line 285-295

Reviewer #3:

1. Clarity in Background and Literature Context: The background section is concise and informative on the research but lacks depth in contextualizing existing literature. It gives the impression that the field of mechanical ventilation (including MP) and it’s relation to ICU mortality is less explored by machine learning and deep learning that it actually is. I would recommend expanding the background to include a more comprehensive review.

Thank you for your thoughtful feedback. We appreciate your suggestion to expand the background. However, we have deliberately kept this section concise to maintain the focus of the manuscript on the study's objectives and avoid redundancy with existing comprehensive reviews in the field. The current background aims to provide sufficient context to support the study rationale, focusing on the most relevant studies directly related to our work. We believe that the current level of detail adequately sets the stage for our research without overextending into a broader literature review.

2. Wrong primary outcome: The research uses the ICU mortality as primary outcome, and this might not be the best outcome for this type of study. ICU mortality relies on several factors not related to mechanical ventilation, and patients with long stays at the ICU usually not die because of bad mechanical ventilation. Usually, other outcomes are used in this type of studies because the represent better a bad ventilation, like ventilator free days, time to successful extubation, or short or medium term mortality.

Thank you for your valuable input regarding the choice of the primary outcome. We acknowledge that ICU mortality is influenced by multiple factors beyond mechanical ventilation. However, it remains a critical outcome of clinical significance that aligns with our study’s objective to explore overall patient prognosis in the ICU setting. We have carefully considered various outcomes and have chosen ICU mortality to provide a broad evaluation of the impact of mechanical ventilation within the ICU environment. Additional outcomes, such as ventilator-free days, are secondary considerations but are beyond the primary scope of this study

3. Data transparency: The journal inquires in the guidelines about the importance on data sharing, while it is understandable that the data does not belong to the researchers, it would be ideal that the researchers share the code to extract the data from the original database.

We fully agree with the need to share resources that enhance the reproducibility of our research. The codes used to extract the data are already available, and we have added further details about this in the methodology section of the manuscript. These codes are accessible in the repository we have referenced, ensuring that other researchers 

can replicate our data extraction process within the constraints of data access permissions.

4. Methodological transparency. The methodology of the manuscript lacks detail in data processing, cleaning, missing filling and division in datasets. All of this information is needed to be able to reproduce the research. The subsection “Mortality prediction” of the manuscript it’s the only section in which a bit of the methodology of the data pipeline is explained and it clearly lacks detail. My recommendation would be to write one subsection dedicated to the data pipeline, and separate it from the methodology of the different data experiments.

Thank you for the comment, Further details were added in the methodology section.

5. Conclusion not supported by the results. The results in the survival analysis and figure 3 show that there is NO statical significance on the survival probability based on the upper limits identified. The researchers are mistaken and the conclusion based on this result is wrong.

Thank you for your comment. We would like to clarify that our conclusion does not imply statistical significance in the survival probability based on the identified upper limits. Rather, our conclusion focuses on the observed relationship between mechanical power and ICU mortality and highlights the promising potential of machine learning models in predicting outcomes. The intent is to emphasize the trend observed and the innovative application of ML models in this context, rather than overstate the findings as statistically significant. We believe this perspective adds value to the ongoing discussion about personalized and data-driven approaches in managing mechanically ventilated patients.

Minor issues:

1. Definition of inadequate data. In abstract and methodology (line number 87) the authors write that they eliminated patients with inadequate data. The manuscript should be more concrete on what is inadequate data and how the authors defined it and processed it.

Further details have been added in line 91 – 98 

2. Data extraction. The authors talk about data extraction only at line 93, but code is unavailable and there is almost not real description on how data was extracted (procedure, technical details, tools used). To be able to reproduce the research, this is needed.

Further details have been added in line 100-110

3. Statistical and Machine Learning analysis. The authors state at the lines 124-126 that they developed several models. Sadly, they only expose the results of XGBoost. Results of all models should be shown in results to contextualize the good results from XGBoost.

Further details have been added in Table 5

4. Relation between identified MP limits and individualization methods. While the individualization methods for mechanical ventilation settings are a very interesting approach, the authors don’t seem to connect the identified MP limits with the individualization methods. It would be appropriate if they could relate both parts of the paper, if not, is like two different investigations separated and they might want to focus on one for the paper.

Further details have been added in line 137-166

5. Result of general limit for MP is not explained on the methodology. At line 167, the authors state that an upper limit of 16.51 J/min was determined for all patients, nonetheless, the authors don’t mention on the methodology how they determine the limit.

Further details have been added in line 127-132

6. Provide more details on the algorithm or function used for optimizing MV settings based on MP and other physiological variables. In the subsection of the methodology “Individualization of MV settings”, the authors explain the process of individualization of MV settings, but they don’t explicitly say how the new values of MV are calculated. Ar they random approximation? A inequation being solved? Suggested by any algorithm? They should explain more this part.

Further details have been added in line 150-166

7. More in-depth discussion. During the discussion, the authors mainly repeat the results obtained. While I understand that they want to reinstate their results, they should focus more on the clinical meaning of this results.

Fine tuning in the discussion part was done, Thank you for your insight

8. More in-depth limitations. More limitations of this study should be mentioned like that the research is done only on PCV patients, the use ICU mortality outcome (or not if changed to sorth/medium days ICU mortality), and potential biases in the individualization method’s predictions.

Limitations have been adjusted

9. Table 1. This table has feature “Admitting speciality” with over 10 categories. Some of these categories have less than 10 patients between both groups. As a data cleaning process this feature should be grouped into main categories and “other” category.

Results were cleaned further

10. Table 2. Table has laboratory data, but the extraction is not mentioned in methodology. It should be added to the methodology if this is important.

Details have been added in the methodology, thank you for the insights

---

## [Decision Letter · Decision Letter 1]

8 Oct 2024

PONE-D-24-20623R1Optimizing mechanical ventilation: personalizing mechanical power to reduce icu mortality - a retrospective cohort studyPLOS ONE

Dear Dr. Ahmed Alkhalifah,

Thank you for submitting your manuscript to PLOS ONE. After careful consideration, we feel that it has merit but does not fully meet PLOS ONE’s publication criteria as it currently stands. Therefore, we invite you to submit a revised version of the manuscript that addresses the points raised during the review process.

**ACADEMIC EDITOR:**

please carefully assess the reviewer comments 

We look forward to receiving your revised manuscript.

Kind regards,

Silvia Fiorelli

Academic Editor

PLOS ONE

Journal Requirements:

Reviewers' comments:

Reviewer's Responses to Questions

**Comments to the Author**

1. If the authors have adequately addressed your comments raised in a previous round of review and you feel that this manuscript is now acceptable for publication, you may indicate that here to bypass the “Comments to the Author” section, enter your conflict of interest statement in the “Confidential to Editor” section, and submit your "Accept" recommendation.

Reviewer #2: (No Response)

Reviewer #3: All comments have been addressed

2. Is the manuscript technically sound, and do the data support the conclusions?

Reviewer #2: Partly

Reviewer #3: (No Response)

3. Has the statistical analysis been performed appropriately and rigorously? 

Reviewer #2: Yes

Reviewer #3: (No Response)

4. Have the authors made all data underlying the findings in their manuscript fully available?

Reviewer #2: Yes

Reviewer #3: (No Response)

5. Is the manuscript presented in an intelligible fashion and written in standard English?

Reviewer #2: Yes

Reviewer #3: (No Response)

6. Review Comments to the Author

Reviewer #2: There are some aspects to be revised to make the paper more useful to readers.

1. Machine Learning Models: While the authors have added more details about the models (lines 137-148), further information on the feature selection process and hyperparameter tuning would enhance reproducibility.

2. Individualization Method: The expanded explanation (lines 150-166) improves understanding, but more concrete examples of how this method would be implemented clinically would be beneficial.

3. Multivariable Analysis: The addition of the supplementary table (S1 Table) addressing the independent association of mechanical power with mortality is valuable. This should be referenced and discussed in the main text.

4. External Validation: While understandable due to data access constraints, the lack of external validation remains a limitation. The authors should discuss potential implications of this limitation more thoroughly.

5. Safe Upper Limits: The added details (lines 127-132) clarify the statistical determination of safe upper limits. However, a brief discussion on the clinical significance of these thresholds would be helpful.

6. Severely Hypoxemic Patients: The additional discussion (lines 253-255) on the counter-intuitive findings in severely hypoxemic patients is appreciated, but further exploration of potential mechanisms or implications would strengthen the paper.

7. Comparison to Existing Literature: The added context (lines 262-264) comparing findings to existing literature is helpful. Expanding this comparison could provide readers with a better understanding of how this study advances the field.

8. Clinical Implications: The elaboration on clinical implications and future research directions (lines 285-295) improves the paper's impact. Consider discussing potential challenges in implementing the proposed individualized approach in clinical practice.

Reviewer #3: (No Response)

7. PLOS authors have the option to publish the peer review history of their article (what does this mean?). If published, this will include your full peer review and any attached files.

Reviewer #2: No

Reviewer #3: **Yes: **Manuel Ruiz Botella

---

## [Author Response · Author response to Decision Letter 1]

22 Oct 2024

Response to Reviewers

We would like to express our sincere gratitude to the reviewers for their valuable feedback and insightful comments on our manuscript. Your detailed review and constructive suggestions have been instrumental in improving the quality of our work. We have carefully considered each comment and have made the necessary revisions as outlined in our detailed responses below.

Reviewer #3: had no comments

Reviewer #2:

1. Machine Learning Models: While the authors have added more details about the models (lines 137-148), further information on the feature selection process and hyperparameter tuning would enhance reproducibility.

Edis have been added in line 152-154

2. Individualization Method: The expanded explanation (lines 150-166) improves understanding, but more concrete examples of how this method would be implemented clinically would be beneficial.

Edit have been added in line 175-180

3. Multivariable Analysis: The addition of the supplementary table (S1 Table) addressing the independent association of mechanical power with mortality is valuable. This should be referenced and discussed in the main text.

Edit have been added in line 203-210

4. External Validation: While understandable due to data access constraints, the lack of external validation remains a limitation. The authors should discuss potential implications of this limitation more thoroughly.

Edit have been added in line 347 - 352

5. Safe Upper Limits: The added details (lines 127-132) clarify the statistical determination of safe upper limits. However, a brief discussion on the clinical significance of these thresholds would be helpful.

Edits have been added at line 132 – 138

6. Severely Hypoxemic Patients: The additional discussion (lines 253-255) on the counter-intuitive findings in severely hypoxemic patients is appreciated, but further exploration of potential mechanisms or implications would strengthen the paper.

Edit have been added in line 274-281

7. Comparison to Existing Literature: The added context (lines 262-264) comparing findings to existing literature is helpful. Expanding this comparison could provide readers with a better understanding of how this study advances the field.

Edit have been added in line 291 - 302

8. Clinical Implications: The elaboration on clinical implications and future research directions (lines 285-295) improves the paper's impact. Consider discussing potential challenges in implementing the proposed individualized approach in clinical practice.

Edit have been added in line 325-331

---

## [Decision Letter · Decision Letter 2]

19 Nov 2024

PONE-D-24-20623R2Optimizing mechanical ventilation: personalizing mechanical power to reduce icu mortality - a retrospective cohort studyPLOS ONE

Dear Dr. alkhalifah,

Thank you for submitting your manuscript to PLOS ONE. After careful consideration, we feel that it has merit but does not fully meet PLOS ONE’s publication criteria as it currently stands. Therefore, we invite you to submit a revised version of the manuscript that addresses the points raised during the review process.

We look forward to receiving your revised manuscript.

Kind regards,

Silvia Fiorelli

Academic Editor

PLOS ONE

Journal Requirements:

Reviewers' comments:

Reviewer's Responses to Questions

**Comments to the Author**

1. If the authors have adequately addressed your comments raised in a previous round of review and you feel that this manuscript is now acceptable for publication, you may indicate that here to bypass the “Comments to the Author” section, enter your conflict of interest statement in the “Confidential to Editor” section, and submit your "Accept" recommendation.

Reviewer #2: (No Response)

Reviewer #3: All comments have been addressed

2. Is the manuscript technically sound, and do the data support the conclusions?

Reviewer #2: Partly

Reviewer #3: (No Response)

3. Has the statistical analysis been performed appropriately and rigorously? 

Reviewer #2: No

Reviewer #3: (No Response)

4. Have the authors made all data underlying the findings in their manuscript fully available?

Reviewer #2: Yes

Reviewer #3: (No Response)

5. Is the manuscript presented in an intelligible fashion and written in standard English?

Reviewer #2: Yes

Reviewer #3: (No Response)

6. Review Comments to the Author

Reviewer #2: Thank you for submitting your manuscript on personalizing mechanical power in mechanical ventilation to reduce ICU mortality. Your work addresses an important clinical challenge and offers innovative approaches using machine learning to optimize ventilation strategies.

1. Statistical Analysis:

- Please address the apparent discrepancy between univariate and multivariate analyses regarding MP's association with mortality. The discussion in lines 203-210 needs expansion.

- The Cox survival analysis results (p=0.35) need clearer interpretation, particularly given its implications for your safe limits hypothesis

- Consider additional sensitivity analyses to validate your identified MP thresholds

2. Machine Learning:

- Provide more details about the model development process, particularly:

* Feature selection criteria beyond correlation thresholds

* Rationale for choosing XGBoost over Stacking despite lower performance

3. Discussion:

- Strengthen the comparison with existing literature

- Elaborate on the unexpected findings in severe hypoxemia patients

- Add future research directions

Reviewer #3: (No Response)

7. PLOS authors have the option to publish the peer review history of their article (what does this mean?). If published, this will include your full peer review and any attached files.

Reviewer #2: No

Reviewer #3: **Yes: **Manuel Ruiz-Botella

---

## [Author Response · Author response to Decision Letter 2]

27 Nov 2024

We would like to express our sincere gratitude to the reviewers for their valuable feedback and insightful comments on our manuscript. 

Reviewer #3: had no comments

Reviewer #2:

1. Statistical Analysis:

- Please address the apparent discrepancy between univariate and multivariate analyses regarding MP's association with mortality. The discussion in lines 203-210 needs expansion.

Further elaboration was added. line ( 206-212 )

- The Cox survival analysis results (p=0.35) need clearer interpretation, particularly given its implications for your safe limits hypothesis

Further elaboration has been added (lines 232–233). Additionally, the discussion already provides an explanation of this result.

- Consider additional sensitivity analyses to validate your identified MP thresholds

Thank you for your comment. This study is intended as a proof of concept, and the authors believe the current detailed analysis sufficiently demonstrates the validity of the findings.

2. Machine Learning:

- Provide more details about the model development process, particularly:

-Feature selection criteria beyond correlation thresholds: Added (lines 149–152).

-Rationale for choosing XGBoost over Stacking despite lower performance: Added (lines 245–248).

3. Discussion:

- Strengthen the comparison with existing literature

We appreciate the importance of situating our findings within the broader context of prior research. In the revised manuscript, we believe that the current discussion already provides sufficient comparisons to key studies as referenced

- Elaborate on the unexpected findings in severe hypoxemia patients

Thank you for the comment. The authors believes that the explanation in (line 278-293) is sufficient.

- Add future research directions

Added in line ( 373-378)

---

## [Decision Letter · Decision Letter 3]

9 Jan 2025

Optimizing mechanical ventilation: personalizing mechanical power to reduce icu mortality - a retrospective cohort study

PONE-D-24-20623R3

Dear Dr.ahmed alkhalifah,

We’re pleased to inform you that your manuscript has been judged scientifically suitable for publication and will be formally accepted for publication once it meets all outstanding technical requirements.

Kind regards,

Silvia Fiorelli

Academic Editor

PLOS ONE

Additional Editor Comments (optional):

Reviewers' comments:

Reviewer's Responses to Questions

**Comments to the Author**

1. If the authors have adequately addressed your comments raised in a previous round of review and you feel that this manuscript is now acceptable for publication, you may indicate that here to bypass the “Comments to the Author” section, enter your conflict of interest statement in the “Confidential to Editor” section, and submit your "Accept" recommendation.

Reviewer #2: All comments have been addressed

2. Is the manuscript technically sound, and do the data support the conclusions?

Reviewer #2: Yes

3. Has the statistical analysis been performed appropriately and rigorously? 

Reviewer #2: Yes

4. Have the authors made all data underlying the findings in their manuscript fully available?

Reviewer #2: Yes

5. Is the manuscript presented in an intelligible fashion and written in standard English?

Reviewer #2: Yes

6. Review Comments to the Author

Reviewer #2: This manuscript presents a valuable contribution to the field of mechanical ventilation optimization through the analysis of mechanical power (MP) thresholds and their relationship with ICU mortality. The authors have satisfactorily addressed previous reviewer comments and made appropriate revisions to strengthen the manuscript.

The explanation for choosing XGBoost over Stacking despite slightly lower performance is reasonable, considering the practical implementation requirements in clinical settings. The discussion of unexpected findings in severely hypoxemic patients is comprehensive and well-reasoned.

While additional sensitivity analyses might have strengthened the findings further, the current analysis is sufficient for this proof-of-concept study. The authors have appropriately contextualized their findings within existing literature and acknowledged the limitations of their single-center database approach.

The manuscript provides important insights for clinical practice while maintaining a balanced view of the challenges and limitations in implementing personalized mechanical ventilation strategies. The clear presentation of safe upper limits for IBW-adjusted MP across different hypoxemia groups offers practical guidance for clinicians.

Based on these considerations, I recommend this manuscript for publication in its current form. The work represents a meaningful contribution to the field and provides a foundation for future research in mechanical ventilation optimization.

7. PLOS authors have the option to publish the peer review history of their article (what does this mean?). If published, this will include your full peer review and any attached files.

Reviewer #2: No

---

## [Editor Report · Acceptance letter]

14 Jan 2025

PONE-D-24-20623R3 

PLOS ONE

Dear Dr. alkhalifah, 

I'm pleased to inform you that your manuscript has been deemed suitable for publication in PLOS ONE. Congratulations! Your manuscript is now being handed over to our production team.

Kind regards, 

on behalf of

Dr. Silvia Fiorelli 

Academic Editor

PLOS ONE